# Influence of Pore-Size/Porosity on Ion Transport and Static BSA Fouling for TiO₂-Covered Nanoporous Alumina Membranes

**Lourdes Gelde [1], Ana Laura Cuevas [2] and Juana Benavente [1],***

1 Departamento de Física Aplicada I, Facultad de Ciencias, Universidad de Málaga, E-29071 Málaga, Spain; geldelourdes@hotmail.com
2 Unidad de Nanotecnología, SCBI Centro, Universidad de Málaga, E-29071 Málaga, Spain; analaura.cuevas@uma.es
* Correspondence: j_benavente@uma.es; Tel.: +34-952-13-1929

**Abstract:** The influence of geometrical parameters (pore radii and porosity) on ion transport through two almost ideal nanoporous alumina membranes (NPAMs) coated with a thin TiO₂ layer by the atomic layer deposition technique (Sf-NPAM/TiO₂ and Ox-NPAM/TiO₂ samples) was analyzed by membrane potential and electrochemical impedance spectroscopy measurements. The results showed the significant effect of pore radii (10 nm for Sf-NPAM/TiO₂ and 13 nm for Ox-NPAM/TiO₂) when compared with porosity (9% and 6%, respectively). Both electrochemical techniques were also used for estimation of protein (bovine serum albumin or BSA) static fouling, and the results seem to indicate deposition of a BSA layer on the Sf-NPAM/TiO₂ fouled membrane surface but pore-wall deposition in the case of the fouled Ox-NPAM/TiO₂ sample. Moreover, a typical and simple optical technique such as light transmission/reflection (wavelength ranging between 0 and 2000 nm) was also used for membrane analysis, showing only slight transmittance differences in the visible region when both clean membranes were compared. However, a rather significant transmittance reduction (~18%) was observed for the fouled Sf-NPAM/TiO₂ sample compared to the fouled Ox-NPAM/TiO₂ sample, and was associated with BSA deposition on the membrane surface, thus supporting the electrochemical analysis results.

**Keywords:** nanoporous TiO₂-covered alumina membranes; BSA fouling, membrane potential; impedance spectroscopy; light transmission/reflection

## 1. Introduction

Nanoporous alumina structures obtained by electrochemical anodization of aluminum foils (two-step method), with pore radius between 10–200 nm and inter-pore distance between 65 and 500 nm (depending on anodization conditions such as electrolyte and voltage), consisting of self-ordered cylindrical aligned channels with narrow pore radius distribution and without tortuosity [1–3], were initially considered as nanofilters for the transport of solutions containing heavy metals ions or as templates for nanotubes and nanowires due to their well-defined structure and high regularity, but other applications such as magnetic storage, solar cells, or photonic crystals have also being studied [4–8]. Besides the high structural regularity of these samples, their strong chemical and thermal resistance are also of great interest when they are used as membranes for controlled release of pharmacologic agents or in food processing due to their stability under the cleaning protocols commonly used to reduce membrane fouling (adsorption/deposition of transported molecules or particles on the membrane structure), which commonly involves the use of chemicals covering a wide range of pH values or high/medium temperature [9–13] and is the main membrane problem in such applications. Moreover, these nanoporous alumina membranes (NPAMs) can also be used in microfluids or as platforms for biochemical/biological sensors and medical devices [14–17], although specific properties such as surface hydrophilic character and biocompatibility or electrical nature are also significant depending on the particular

application. Consequently, possible easy membrane surface modification able to confer the membrane optimal geometrical and material characteristics for a specific application is of great interest. Different techniques such as dip coating, grafting, chemical vapor deposition, plasma polymerization, or atomic layer deposition have already been proposed for NPAM surface functionalization/modification [18–20]. In particular, the atomic layer deposition (ALD) technique is a suitable manner of modifying both membrane pore size and surface physicochemical characteristics (charge, hydrophilicity, or biocompatibility) by covering external and internal (pore-wall) surfaces of NPAMs with a homogeneous layer of a selected material (nitrides, sulfides, or metal oxides such as $SiO_2$, $ZnO$, $TiO_2$, $Fe_2O_3$, or even $Al_2O_3$, when only geometrical parameters have to be modified), providing to these nanoporous alumina-based membranes (NPA-bMs) desired geometrical and material properties [20,21]. In fact, the ALD technique is especially appropriate for modification of membranes with high aspect ratios (that is, pore length/pore diameter > 2000), and it also allows surface modification with two or more layers from different materials [22].

As indicated above, a significant problem associated with the use of planar NPA-bMs in the separation/transport of solutions containing proteins or macromolecules is membrane fouling, because it can significantly reduce the effectiveness of the process, necessitating the development of efficient cleaning procedures [9]. Streaming potential measurements have traditionally been performed for determination and analysis of membrane fouling, although the pressure difference needed for such measurements could not be appropriate for fragile samples as is the case with the NPA-bMs indicated above. Other kinds of electrochemical experiments such as membrane potential ($\Delta\Phi_{mbr}$) or electrochemical impedance spectroscopy (EIS) have also been used for membrane fouling estimation [23–27], but different and specific optical techniques (ellipsometry spectroscopy or optical coherence tomography) have also been proposed [28,29]. On the other hand, it was already reported that modification with $TiO_2$ nanoparticles of different types of membranes for diverse applications (membrane bioreactors, oily emulsion filtration, etc.) seems to reduce the fouling effect independently of membrane material and $TiO_2$ particle immobilization procedure (entrapped, deposited, or deep coating) [30–32]. In this context, the increase in the hydrophilic character of $TiO_2$-modified membranes (polymeric and ceramic) has been indicated as one of the key factors for membrane fouling reduction caused by filtration of oily wastewater or BSA static fouling, although other factors such as $TiO_2$ surface charge, which depends on solution pH and salt concentration, have also been considered [33–35]. Moreover, the biocompatibility and corrosion resistance of $TiO_2$, of great importance for its use in dental implants, has already been reported [21].

In this work, we considered the influence of geometrical parameters on both ion transport and static protein (bovine serum-albumin or BSA) fouling for two nanoporous alumina-based membranes obtained by $TiO_2$-layer coverage using the ALD technique, with similar thickness but different pore size and porosity, by analyzing and comparing membrane potentials and EIS diagrams determined for clean and fouled membranes in order to obtain qualitative and quantitative information on the BSA fouling process. According to the results, the fouling mechanism seems to be strongly dependent on pore radius when compared to the porosity effect, although interfacial effects might affect particular quantification. However, due to the high transparency of the studied membranes, optical analysis using a typical and simple characterization technique such as light transmission/reflection was also used to confirm the membrane fouling mechanism established from electrochemical measurements.

## 2. Materials and Methods

### 2.1. Materials

The nanoporous alumina membranes (NPAMs) used as supports in this study were synthesized by the electrochemical two-step anodization method using high purity aluminum discs (Al 99.999%, Goodfellow (UK); 0.5 mm thickness). Two different aqueous electrolyte solutions and anodization voltages were used in order to have membranes

with different pore size and interpore distance: 0.3 M solution of sulfuric acid and applied voltage of 25 V for the Sf-NPAM sample, and 0.3 M solution of oxalic acid and 40 V in the case of the Ox-NPAM sample; detailed information on this process is given in the literature [1,3,35]. In Supplementary Information, scanning electron microscope (SEM) pictures of both membrane surfaces are presented as Figure S1.

Modification of Sf-NPAM and Ox-NPAM surfaces, both external and internal (pore-wall), with a thin $TiO_2$ layer coated by the atomic layer deposition (ALD) technique was performed in a Savannah 100 thermal ALD reactor from Cambridge Nanotech (Waltham, MA, USA), using high purity argon as carrier gas. Two precursors, titanium tetraiso-propoxide (metal precursor) at 75 °C and water (for substrate functionalization and as oxidant agent) at 60 °C, were used for deposition of the conformal coating, as explained in a previous work [20]. These nanoporous alumina-based membranes will hereafter be called Sf-NPAM/$TiO_2$ and Ox-NPAM/$TiO_2$, respectively. Almost total $TiO_2$ coverage of both alumina supports was already determined in a previous paper [36] by the XPS technique, and the atomic concentration percentage (A.C. %) of the different elements found on the surfaces of each membrane is shown in Supplementary Information (Table S1); only superficial slight differences associated with manufacture/environmental contamination (sulfur detection for Sf-NPAM/$TiO_2$ or excess carbon in the case of the Ox-NPAM/$TiO_2$ sample) were determined. The thickness of the $TiO_2$ layer was also determined (from depth-profile XPS analysis [36]), obtaining a value of ~6 nm, which agrees quite well with published results [21]. Figure 1 shows SEM surface micrographs of both NPAM/$TiO_2$ samples. The following values for average pore-radii ($r_p$), interpore distance ($D_{int}$) and porosity ($\Theta = (2\pi/\sqrt{3})(r_p/D_{int})^2$) [1]), were determined by surface analysis of SEM micrographs [36]: $<r_p> = 10$ nm, $<D_{int}> = 65$ nm, and $<\Theta> = 9\%$ for the Sf-NPAM/$TiO_2$ sample, and $<r_p> = 13$ nm, $<D_{int}> = 105$ nm, and $<\Theta> = 6\%$ for the Ox-NPAM/$TiO_2$ sample, with both membranes having a thickness of ~65 μm. Taking into account geometrical parameter values, rather similar theoretical hydraulic permeability was determined for both $TiO_2$-covered membranes [9]: $L_H^{Sf-NPAM/TiO2} = 1.92 \times 10^{-12}$ m/s.Pa and $L_H^{Ox-NPAM/TiO2} = 2.17 \times 10^{-12}$ m/s.Pa. It should be mentioned that both NPAMs and NPAM/$TiO_2$ samples were obtained in the Unidad de Membranas Nanoporosas, Universidad de Oviedo (Spain) and kindly submitted by Prof. V. de la Prida and Dr. V. Vega.

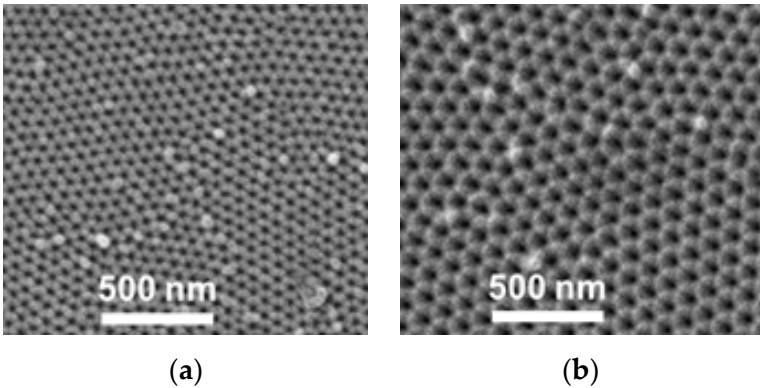

(**a**)  (**b**)

**Figure 1.** SEM micrograph of: (**a**) Sf-NPAM/$TiO_2$ membrane surface and (**b**) Ox-NPAM/$TiO_2$ membrane surface (values and reference in text).

Membranes were statically fouled with the protein bovine serum albumin (BSA), which had the following characteristic parameters: 66.5 kDa molecular weight and Stokes radius of 3.48 nm. Membrane fouling was performed by mounting each sample in the dead-end electrochemical cell shown in Supplementary Information (Figure S2), filling one half-cell with a BSA aqueous solution of 5 g/L and the other half-cell with distilled water for 24 h, after which the cell was emptied and the membrane surface softly cleaned with distilled water. These samples will hereafter be called Sf-NPAM/$TiO_2$(f) and Ox-NPAM/$TiO_2$(f).

## 2.2. Electrochemical Characterization

Membrane potentials ($\Delta\Phi_{mbr}$) and equilibrium electrical potential differences between the 2 NaCl solutions of different concentrations (Cf = 0.01 M and Cv ranging between 0.002 M and 0.1 M) at both membrane sides were determined by measurements performed in the dead-end test cell already indicated (Supplementary Information, Figure S2), with the 2 Ag/AgCl reversible electrodes (to Cl$^-$ ion) connected to a digital voltmeter (Yokohama 7552, 1 G$\Omega$ input resistance) and magnetic stirrers placed in the bottom of each cell working at a stirring rate of 540 rpm (to minimize concentration polarization at the membrane surfaces [37]). Because measured values ($\Delta$E) included two different contributions, electrode potential ($\Delta\Phi_{elect}$) and membrane potentials ($\Delta\Phi_{mbr}$), the latter values were obtained by subtracting from each $\Delta$E value the corresponding $\Delta\Phi_{elect}$ (which depended on electrolyte concentration, $\Delta\Phi_{elect} = (RT/zF)\ln(C_v/C_f)$, where R and F are gas and Faraday constants, respectively, while T represents the temperature of the system); hence: $\Delta\Phi_{mbr} = \Delta E - \Delta\Phi_{elect}$ [38]. Measurements were performed at standard pH (5.8 $\pm$ 0.3) and room temperature (25 $\pm$ 2) °C.

Electrochemical impedance spectroscopy (EIS) is an alternating current (a.c.) technique used for electrical characterization of membranes in "working condition", that is, in contact with electrolyte solutions. EIS allows, under certain conditions, the estimation of different contributions: membrane, electrolyte solution, and membrane/solution interface, by using equivalent circuits as models [39,40]. The impedance (Z) is a complex number, with real ($Z_{real}$) and imaginary ($Z_{img}$) parts ($Z = Z_{real} + j\,Z_{img}$) that are related to both the transport of charge across the membrane and charge storage, through its electrical resistance (R) and capacitance (C) [39]. The analysis of the impedance data was performed by using the Nyquist plot in the complex plane ($-Z_{img}$ versus $Z_{real}$) [39]. EIS measurements were carried out with the membranes in the electrochemical test cell by connecting the electrodes to a Frequency Response Analyzer (FRA, Solartron 1260, Hamshire, UK) using a maximum voltage of 0.01 V and an interval of frequency ranging between 1 Hz and $10^7$ Hz, in open circuit mode. The ZView 2 data analysis program (Scribner, Southern Pines, NC, USA) was used for determination of electrical parameters, and EIS data were corrected by the influence of connecting cables and other parasite capacitances.

## 2.3. Optical Measurements

Optical characterization of clean and fouled samples was performed by analyzing transmittance and reflectance spectra, which were recorded with a Varian Cary 5000 spectrophotometer (Agilent Technologies, Santa Clara, CA, USA) provided with an integrating sphere of Spectralon for a wavelength interval of 0–2000 nm.

## 3. Results and Discussion

### 3.1. Electrochemical Characterizations

Ion diffusive transport across membranes, which depends on membrane material electrical characteristics and pore size, as well as the electrolyte and solution concentrations, can be studied by analyzing membranes' potential values. This analysis allows the estimation of membrane effective fixed charge concentration ($X_{ef}$) and ion transport numbers (ti), or fraction of the total current transported through the membrane by one ion ($t_i = I_i/I_T$) [38]; for an ideal positively charged membrane (anion exchanger), t$-$ = 1 and t+ = 0, that is, only negative charges can pass through the membrane, while for cation exchangers or ideal negatively charged membranes, t+ = 1 and t$-$ = 0. Effective fixed charge controls Donnan (or interfacial) potential ($\Delta\emptyset_{Don}$), while values of ion transport numbers in the membrane also depend on pore size and are directly related to diffusion potential ($\Delta\varphi_{dif}$). The expressions for Donnan and diffusion potentials are [38]:

$$\text{Donnan potential, } \Delta\emptyset_{Don} = (RT/F)\ln[(wX_{ef}/2C) + [(wX_{ef}/2C)^2 + 1]^{1/2}]] \tag{1}$$

$$\text{Diffusion potential, } \Delta\varphi_{dif} = -(RT/F)[(t- - t_+)]\ln(C_v/C_f) = (RT/F)[(1 - 2t-)]\ln(C_v/C_f) \tag{2}$$

with w = −1/+1 for negatively/positively charged membranes, while $C_i$ indicates solution concentration (i = v for variable solution concentration and i = f for fixed solution concentration), and R, F, and T have been previously identified. According to the Teorell-Meyer-Sievers model [41,42], membrane potential consists of the sum of two Donnan potentials (one for each membrane/solution interface) and the diffusion potential in the membrane.

Figure 2 shows a scheme of ion transport through two porous membranes of the same material (similar positive charge) but different pore size and porosity, as is the case for the Sf-NPAM/TiO$_2$ and Ox-NPAM/TiO$_2$ samples. Figure 2a shows a membrane with small pore size, which behaves as an almost ideal anion-exchanger, avoiding the pass of practically all cations (membrane co-ions), but the increase of pore size favors the inclusion of electrolyte solution into the pores as shown in Figure 2b, which raises the transport of cations and increases $t_+$ value. Because ion transport numbers depend not only on membrane charge but also electrolyte electrochemical parameters, membrane ionic permselectivity ($PS_i$), which represents the relative variation of counter ions in the membrane with respect to their values in solution, is usually considered; then, for positively charged membranes and NaCl solutions, $PS_{Cl^-}$, is expressed as [43]: $PS_{Cl^-} = (t_{Cl^-} - t^o_{Cl})/t^o_{Na^+}$, where $t_{Cl^-}$ is the transport number of the counter-anion in the membrane pores, while $t^o_{Cl^-}$ and $t^o_{Na^+}$ represent the transport numbers of the anion and cation, respectively, in the NaCl solution.

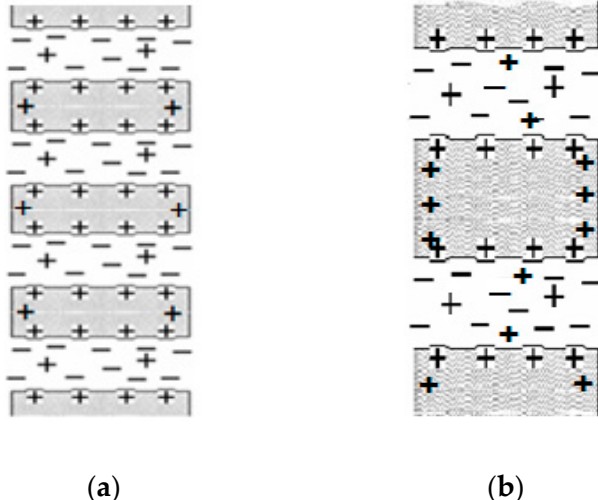

(**a**)          (**b**)

**Figure 2.** Scheme of ion transport through two positively charged nanoporous membranes with the same thickness and material but different pore size and porosity. (**a**) Pore charge practically rejects all co-ion favoring counter-ion transport in membranes with reduced pore radii; (**b**) higher pore size favors co-ion inclusion, increasing co-ion transport number.

Figure 3a shows the dependence of membrane potential values on solution concentration ratio for clean Sf-NPAM/TiO$_2$ and Ox-NPAM/TiO$_2$ membranes, but concentration dependence for solution diffusion potential (dashed line) obtained using solution transport numbers in Equation (2) ($t_{Na}+^o$ and $t_{Cl}-^o$ [44], dashed line]; theoretical $\Delta\Phi_{mbr}$ values for an ideal anion-exchanger membrane (t− = 1, solid line) are also represented for comparison. As can be observed, membrane potential values for the sample with lower pore radii, Sf-NPAM/TiO$_2$, hardly differ from those corresponding to the ideal positively charged membrane, but reduced $\Delta\Phi_{mbr}$ values were obtained for the Ox-NPAM/TiO$_2$ membrane (according to Figure 2 explanations), as the difference is more significant at high concentration ratio values as the result of a fixed charge screening effect. On the other hand, dependence of $\Delta\Phi_{mbr}$ values on the electrolyte solution can be observed in Figure 3b, where a comparison of data points for the Sf-NPAM/TiO$_2$ membrane using KCl and CaCl$_2$ is shown. The combination of both membrane geometry and surface material can be observed in Supplementary Information (Figure S3), where a comparison of membrane

potential values measured with KCl solutions for Sf-NPAM, Ox-NPAM, Sf-NPAM/TiO$_2$, and Ox-NPAM/TiO$_2$ samples is presented [36], while the effect of pore size, porosity, and structure asymmetry for NPAMs has already been reported [35,45].

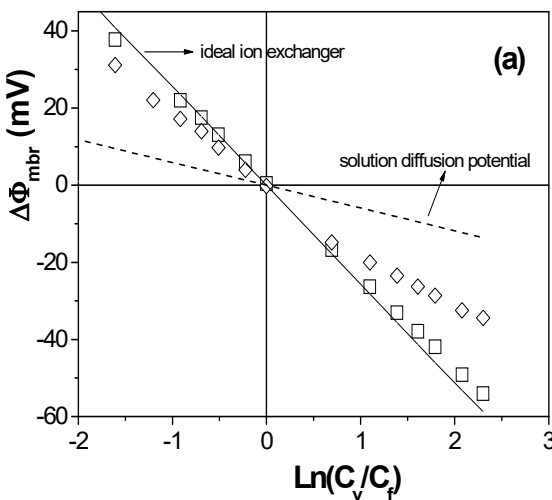

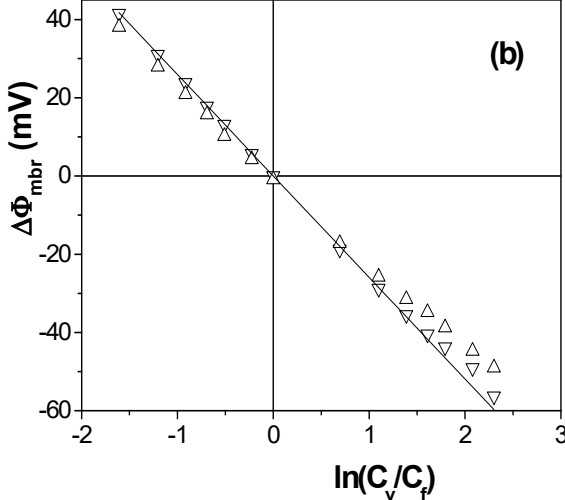

**Figure 3.** Membrane potential as a function of solution ratio for: (**a**) Sf-NPAM/TiO$_2$ membrane (□), Ox-NPAM/TiO$_2$ membrane (◊) solution diffusion potential (dashed line) for NaCl electrolyte, and (solid line) ideal positively charged membrane (t− = 1); (**b**) Sf-NPAM/TiO$_2$ membrane and KCl solutions (△), CaCl$_2$ solutions (▽).

The fit of the experimental values can be performed taking into account both Donnan and diffusion potentials as well as membrane electroneutrality, and it allows the determination of the effective membrane fixed charge concentration ($X_{ef}$), anion/cation diffusion coefficient ratio ($D_{Cl-}/D_{Na+}$, because ($t_{Cl-}/t_{Na+}$) = ($D_{Cl-}/D_{Na+}$) [44]) or anion permselectivity, as has already been explained in previous papers [46–48]. The following values were determined: $X_{ef}$ = + 0.015 M, $D_{Cl-}/D_{Na+}$ = 3.74, and $P_{Cl-}$ = 45.2% for Sf-NPAM/TiO$_2$ membrane, but $X_{ef}$ = + 0.012 M, $D_{Cl-}/D_{Na+}$ = 2.47, and $P_{Cl-}$ = 25.3% for Ox-NPAM/TiO$_2$ membrane. These results show the reduction in pore size increases for ionic diffusion coefficient ratio, which tends to a value more similar to that for NaCl solution, ($D_{Cl-}/D_{Na+}$)$^o$ = 1.60), and logically for anionic permselectivity, with a reduction of 44%, both associated with higher electrolyte content into the pores of the Ox-NPAM/TiO$_2$ membrane. Figure 4 shows a comparison of experimental and fitted values as well as

the separate contribution of Donnan and diffusion potentials as a function of variable concentration $C_v$ for Sf-NPAM/TiO$_2$ and Ox-NPAM/TiO$_2$ membranes, where the agreement between experimental and theoretical values support the suitability of these results. On the other hand, a correlation between effective fixed charge concentration ($X_{ef}$) and surface charge density ($\sigma_s$) for cylindrical pores was already proposed by the following expression [49]: $\sigma_s = (X_{ef}.F.r_p/2)$, where F represents Faraday constant. Surface charge density values for the studied membranes were 7.2 mC/m$^2$ for the Sf-NPAM/TiO$_2$ sample and 7.5 mC/m$^2$ for the Ox-NPAM/TiO$_2$ sample. Because the electrical charge exhibited by most membranes in contact with aqueous solutions (polar medium) corresponds to the adsorption of charged species from solution on the pore walls and/or dissociation of specific ionic groups, the similarity obtained for surface charge density values seems to be in concordance with the similitude in membrane surface material determined from XPS analysis.

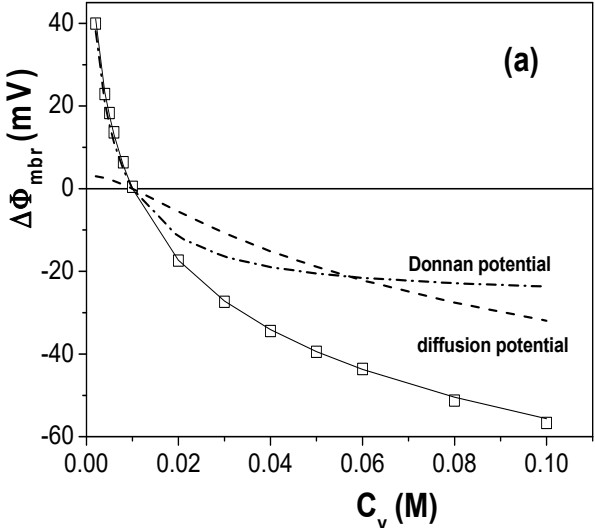

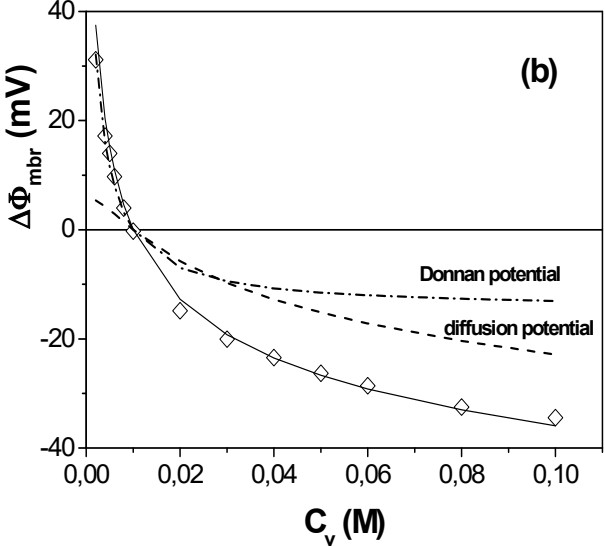

**Figure 4.** Membrane potential as a function of variable solution $C_v$. (**a**) Sf-NPAM/TiO$_2$ membrane (□) experimental values and (**b**) Ox-NPAM/TiO$_2$ membrane (◇) experimental values, fitted values (solid line), diffusion potential contribution (dashed line) and Donnan potential contribution (dashed dot line). Fitted values: (**a**) $X_{ef}$ = +0.015 M, $t_{Cl}-/t_{Na}+$ = 3.74; (**b**) $X_{ef}$ = +0.012 M, $t_{Cl}-/t_{Na}+$ = 2.47.

Another electrochemical technique used for characterization of membranes in working conditions is electrochemical impedance spectroscopy (EIS). This technique gives quantitative and/or qualitative information related to charge movement or charge adsorption by means of the electrical resistance or capacitance (equivalent capacitance in case of non-homogeneous systems [39]) respectively, which can be significantly affected by membrane structure and material characteristics [50,51]. Qualitative information is obtained by comparing impedance diagrams for membranes from the same material but different geometry, while characteristic parameter values are estimated by the fitting of experimental data to adequate equivalent circuits [39,40]. A comparison of impedance diagrams for clean Sf-NPAM/TiO$_2$ and Ox-NPAM/TiO$_2$ membranes measured in the system: electrode//electrolyte solution (C)/membrane/electrolyte solution (C)//electrode, with C = 0.002 M NaCl, is shown in Figure 5. The Nyquist plot ($-Z_{img}$ vs. $Z_{real}$) shows significant differences between both membranes, because the impedance diagram for Sf-NPAM/TiO$_2$ sample exhibits two differentiated relaxations that correspond to the separated contributions of membrane (m) and electrolyte solution (e) (see insert in Figure 5 where this part of the diagram is enlarged), while a unique relaxation process is observed in the diagram for the Ox-NPAM/TiO$_2$ sample, which corresponds to the "membrane plus electrolyte" contribution. As already reported, differences in impedance diagrams can depend on both membrane structure/geometry and material hydrophilic/hydrophobic character, because membrane and electrolyte separated impedance contributions have been obtained for composite reverse osmosis and even nanofiltration membranes with a two-layer structure (a denser thin active layer and a porous support) [48,52,53], while only a relaxation process was observed for wider pore size ultrafiltration/microfiltration membranes [54,55] and also for nanoporous regenerated cellulose membranes (2 kDa cut off, which should correspond to a pore size ~ 3.0 nm) associated with high water (aqueous solution) adsorption due to the hydrophilic character of this membrane material [48]. On the other hand, the analysis of impedance diagrams allows us to obtain the following values for the electrical resistance of Sf-NPAM/TiO$_2$ and Ox-NPAM/TiO$_2$ membranes: $R^{Sf-NPAM/TiO2}$ = 4510 ohm and $R^{Ox-NPAM/TiO2}$ = 1665 ohm, although this latter value was determined by subtraction of electrolyte electrical resistance ($R_e$) from that determined for the Ox-NPAM/TO$_2$ membrane plus electrolyte system ($R_{me}$); hence: ($R_m = R_{me} - R_e$), that is, $R_m$ was obtained from two different measurements that may slightly affect the real value. In any case, due to the similar material of both NPAM+TO$_2$ samples, EIS analysis for clean membranes is concordant with that previously performed by membrane potential results with respect to pore radii influence.

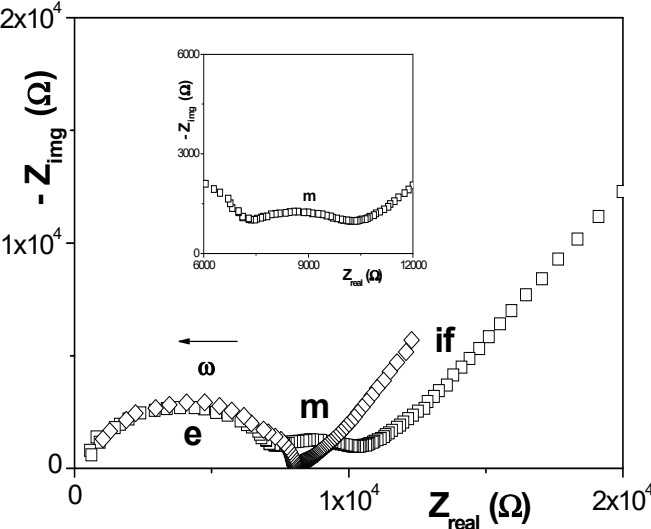

**Figure 5.** Nyquist plots for (☐) Sf-NPAM/TiO$_2$ membrane and (◇) Ox-NPAM/TiO$_2$ membrane. (e) Electrolyte solution; (m) membrane; (if) interface; ω: angular frequency and values increase tendency.

The BSA fouling effect on membrane potentials for the Ox-NPAM/TiO$_2$(f) sample is shown in Figure 6a where, for comparison, $\Delta\Phi_{mbr}$ values for the clean Ox-NPAM/TiO$_2$ are also indicated. Both membranes show rather similar values at low concentrations (interfacial or Donnan potential contribution) but differ at high concentrations, which is an indication of differences in ion diffusion between both samples that might be caused by pore size reduction due to BSA deposition on pore walls (although surface deposition might also exist). Fitted values of characteristic parameters were $X_{ef}$ = +0.014 M, $D_{Cl-}/D_{Na-}$ = 5.4, and $P_{Cl-}$ = 58.2%, as these two latter values were much higher than those determined for the clean membrane. Figure 6b shows the comparison between experimental and fitted values depending on variable solution $C_v$, as well as the separate contributions of diffusion and Donnan potentials; this second contribution does not differ significantly from that in Figure 3b for clean membrane at high $C_v$ values.

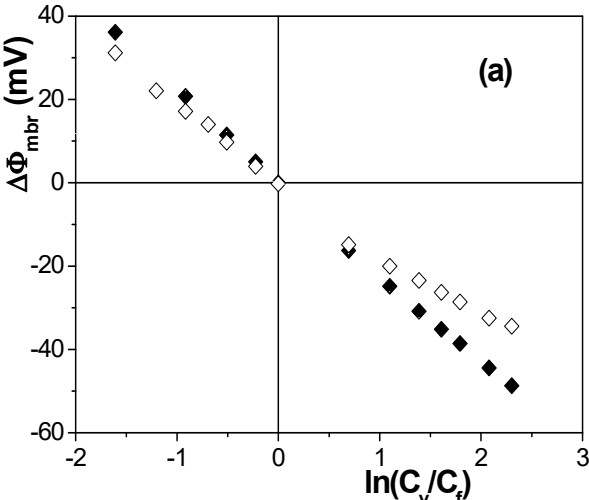

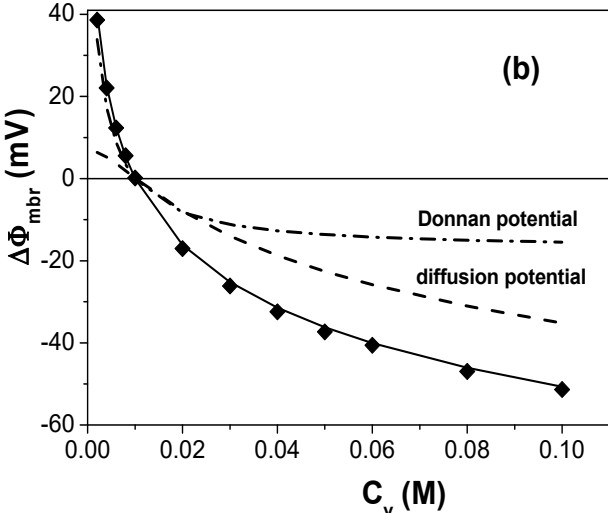

**Figure 6.** (**a**) Comparison of membrane potential with solution concentration ratio for Ox-NPAM/TiO$_2$ (◊) and Ox-NPAM/TiO$_2$(f) (♦) membranes. (**b**) Comparison of experimental and calculated (solid line) values, Donnan (dashed-dot line), and diffusion (dashed line) potential contributions for Ox-NPAM/TiO$_2$(f) membrane.

However, the protein fouling effect or mechanism exhibited by the Sf-NPAM/TiO$_2$(f) membrane seems to be different according to results shown in Figure 7a, where a similar concentration ratio dependence for fouled and clean samples can be observed, but the results obtained for the Sf-NPAM/TiO$_2$(f) membrane present an almost constant shift to lower $\Delta\Phi_{mbr}$ values. This result could indicate the presence of a BSA layer on the membrane surface, rather than pore-wall deposition (which could be related with a "static" fouling nature because any hydrostatic pressure is applied to the BSA solution), giving rise to a kind of "composite" or two-layer membrane, as can be deduced from results shown in Figure 6b, where $\Delta\Phi_{mbr}$ values for the fouled sample after subtraction of that determined at $C_v = C_f = 0.01$ M are compared with those determined for the clean Sf-NPAM/TiO$_2$ membrane, and only small differences exist between them. Figure 7b also shows the data points obtained when $\Delta\Phi_{mbr}$ values for the clean membrane are subtracted from those corresponding to the fouled sample. In this context, the difficulty of obtaining quantitative information from these results should be pointed out because they do not correspond to similar interfacial situations, and a compact BSA layer on membrane surface may also affect the concentration gradients.

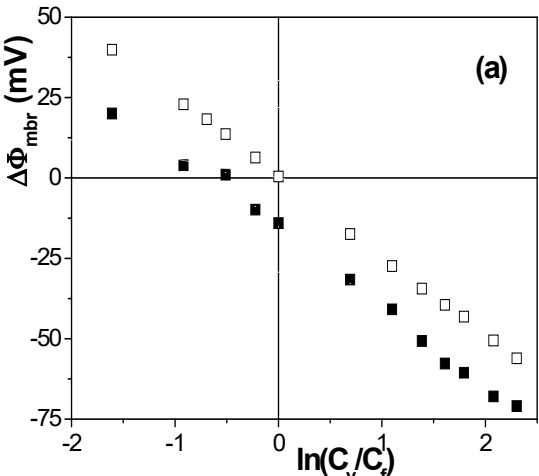

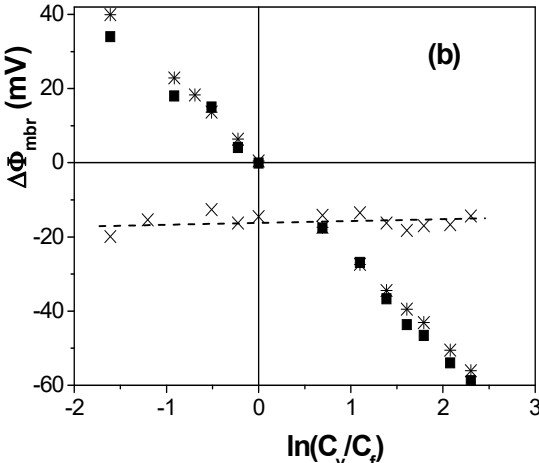

**Figure 7.** (**a**) Comparison of membrane potential with solution concentration ratio for Ox-NPAM/TiO$_2$ (◊) and Ox-NPAM/TiO$_2$(f) (♦) membranes. (**b**) Comparison of experimental and calculated (solid line) values, Donnan (dashed-dot line), and diffusion (dashed line) potential contributions for Ox-NPAM/TiO$_2$(f) membrane.

EIS measurements have also been used in the study of porous membrane fouling by comparing impedance plots obtained with clean and fouled samples [56,57], as shown in Figure 8 for Ox-NPAM/TiO$_2$ and Ox-NPAM/TiO$_2$(f) membranes. Analysis of these results has permitted the estimation of around 10% increase in the electrical resistance of the fouled sample. However, EIS measurements of the Sf-NPAM/TiO$_2$(f) membrane have not been performed due to enormous data fluctuations, which were attributed to BSA deposited layer instability associated with a cyclic alternating current effect.

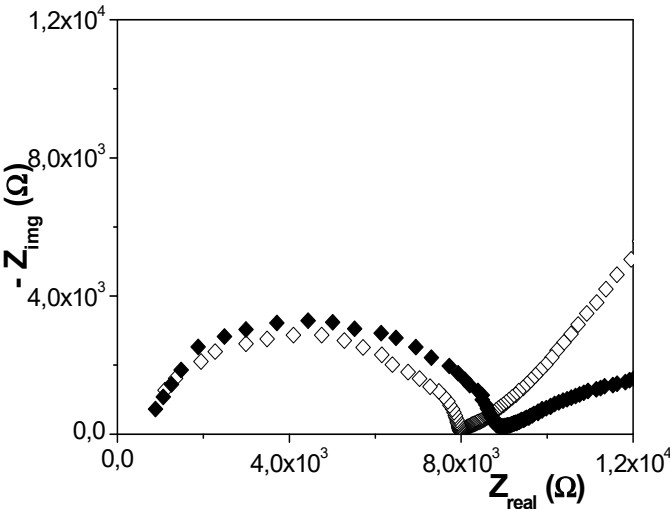

**Figure 8.** Comparison of Nyquist plot for: (◊) Ox-NPAM/TiO$_2$ membrane and (♦) Ox-NPAM/TiO$_2$(f) membrane (C = 0.002 M NaCl).

### 3.2. Optical Characterization

Optical techniques (light transmission and spectroscopic ellipsometry), which are of great interest due to their non-destructive character, have already demonstrated their suitability for characterization of nanoporous alumina membranes with different pore sizes and porosity (or even asymmetric structure) or human serum albumin adsorption on alumina supports [35,58]. In this context, depending on geometrical parameters (pore-size/porosity) or surface material, differences in light transmission through nanoporous alumina-based membranes are expected. In fact, differences in transmission spectra through membranes obtained by coating Sf-NPAM or Ox-NPAM samples with a layer of different metal oxides have already been analyzed [20,22]. Figure 9a shows a comparison of transmission spectra for two samples obtained by coating the Sf-NPAM support with a layer of Fe$_2$O$_3$ or Al$_2$O$_3$ (Sf-NPAM/Fe$_2$O$_3$ or Sf-NPAM/Al$_2$O$_3$ samples, with similar pore-size, porosity, and thickness values than Sf-NPAM/TiO$_2$), and clear differences in the visible region can be observed due to the different surface material (although both membranes show high and similar transparency in the near infrared region); moreover, transmission for another membrane, obtained by coating the Ox-NPAM support with a SiO$_2$ layer (Ox-NPAM/SiO$_2$ sample), of different pore-size, porosity, and surface material than the Sf-NPAM/Fe$_2$O$_3$ and Sf-NPAM/Al$_2$O$_3$ samples, was also measured, and transmission differences in both visible and near infrared regions were obtained. These previous results seem to support the possibility of using light transmission spectra for changes associated with membrane fouling for this kind of sample.

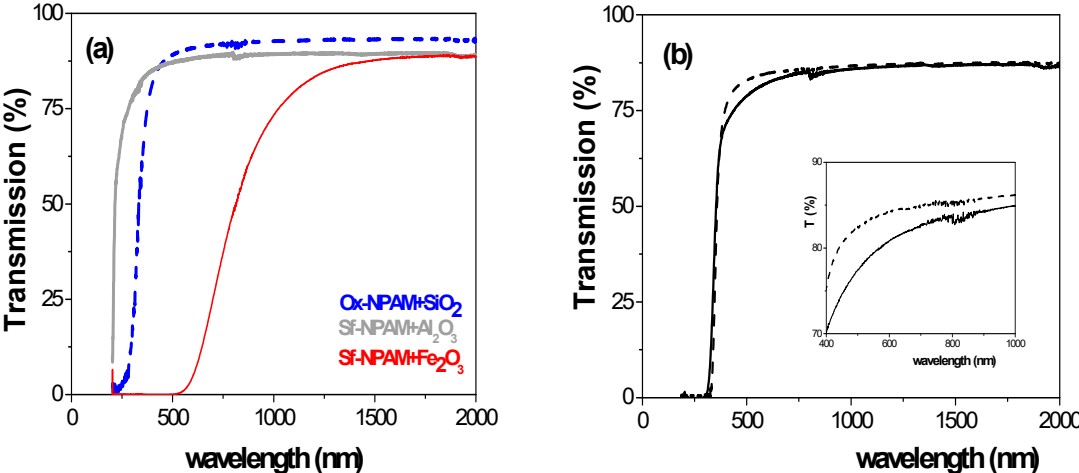

**Figure 9.** Light transmission spectra for (**a**) Sf-NPAM/Al$_2$O$_3$ (solid gray line), Sf-NPAM+Fe$_2$O$_3$ (solid red line), and Ox-NPAM/SiO2 (solid blue line) samples; (**b**) Sf-NPAM/TiO$_2$ (solid black line) and Ox-NPAM/TiO$_2$ (dashed black line) samples.

Then, optical characterization of both clean membranes was firstly considered and Figure 9b shows a comparison of the transmittance spectra, where the high transparency of both membranes (>70% in the visible region) and around 88% for near infrared region (NIR) can be seen. The similarity in light transmission exhibited by both membranes is significant, taking into account that they have different geometrical parameters. This may be due to a kind of pore-size/porosity compensation, as the sample with lower pore radii presents higher porosity and the free volume of both membranes is rather similar (Ox-NPAM/TiO$_2$/Sf-NPAM/TiO$_2$ free volume ratio = 1.13), as indirectly established by the similarity of hydraulic permeability values already indicated. On the other hand, transmittance analysis also provides an indirect confirmation of the adequate TiO$_2$-coverage of the nanoporous alumina support by the estimation of membrane band gap values, which are between 310 nm and 323 nm and agree with reported values for TiO$_2$ [59], differing significantly from the band gap for Al$_2$O$_3$ materials (around 200 nm [60], and logically from Fe$_2$O$_3$ as can be seen in Figure 9a); this point agrees with the almost total TiO$_2$-coverage of the alumina support determined by the XPS technique [36] already mentioned (Table S1 in Supplementary Information).

The effect of membrane BSA fouling on light transmission spectra can be observed in Figure 10. In the case of the Ox-NPAM/TiO$_2$(f) membrane (Figure 10a), a certain wavelength-dependent reduction in the visible region was determined (9.0% at 600 nm), but almost no change is observed in the near infrared region. This small effect of BSA fouling on light transmission might support protein deposition on pore walls as the main fouling mechanism determined from electrochemical measurements, due to the reduced free volume of studied membranes and taking into account the high transparency exhibited by their solid structure. However, an important reduction in light transmission for the whole wavelength range can be observed in Figure 10b for the Sf-NPAM/TiO$_2$(f) sample when compared with the Sf-NPAM/TiO$_2$ sample; in particular, a reduction of around 18.0% at 600 nm and of 7.0% in the NIR (at 1600 nm) were obtained. This reduction in transmission through the fouled membrane could be an indication of the presence of a BSA layer on the membrane surface, which would modify the high transparency of membrane material indicated above; this fact would be in agreement with membrane potential analysis previously performed. Consequently, light transmission measurements for membranes made of highly transparent materials could be used as a way to determine fouling mechanisms by comparing spectra for clean and fouled samples.

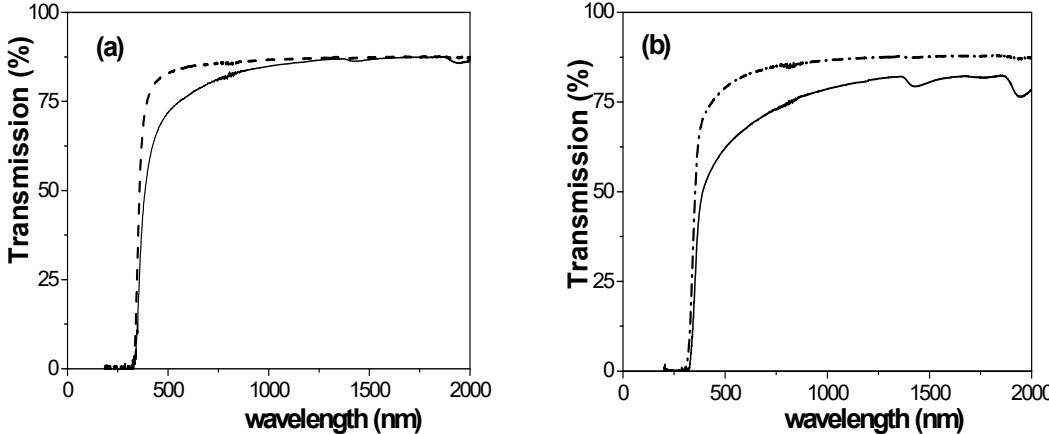

**Figure 10.** Comparison of light transmission spectra for (**a**) clean Ox-NPAM/TiO$_2$ (dashed line) and fouled Ox-NPAM/TiO$_2$(f) (solid line) membranes; (**b**) clean Sf-NPAM/TiO$_2$ (dashed dot line) and fouled Sf-NPAM/TiO$_2$(f) (solid line) membranes.

Reflection spectra (R%) for clean and fouled membranes were also measured, and the corresponding absorption curve (A%) was obtained: A (%) = 100 − T (%) − R (%). Figure 11 shows a comparison of wavelength dependence of absorption spectra for both BSA fouled samples; for comparison, absorption spectra for both clean samples are shown in the inlet (for wavelengths between 500 nm and 2000 nm). Differences between both fouled samples can be observed in the visible region, with a significant adsorption reduction, but more closely similar and constant values were obtained for both clean samples.

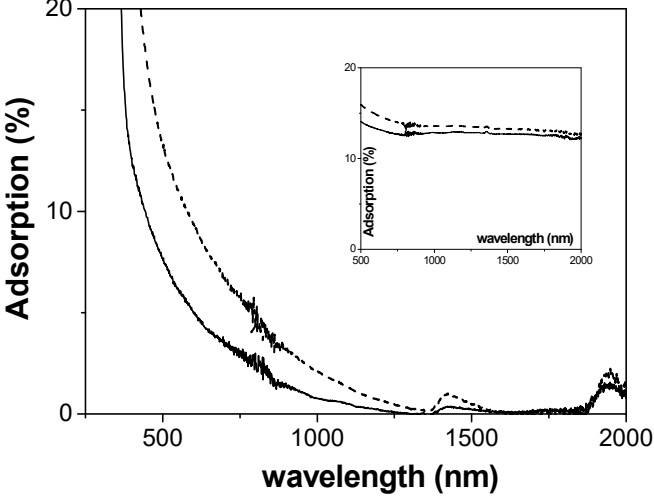

**Figure 11.** Comparison of light absorption spectra for Sf-NPAM/TiO$_2$ fouled membrane (solid line) and Ox-NPAM/TiO$_2$ fouled membrane (solid line). Insert: comparison of absorption spectra for Sf-NPAM (solid line) and Sf-NPAM (dashed line) samples.

## 4. Conclusions

The influence of pore radius on ion transport through nanoporous alumina-based membranes with almost ideal porous structure (TiO$_2$-coated alumina nanoporous supports with different pore radii and porosity) was analyzed by membrane potential and electrochemical impedance spectroscopy measurements. These results show the significant effect of pore size. Moreover, both electrochemical techniques can also be used for quantitative/qualitative information on membrane fouling dependent on pore size: a BSA layer on the membrane surface for the sample with lower pore radii, but pore-wall deposition in the case of the membrane with wider pore size. On the other hand, the high transparency of a membrane material enables the use of a common optical technique,

light transmission/reflection, to obtain information on the fouling effect or mechanism. Optical results appear to confirm the findings of electrochemical analysis and also provide information on other membrane material characteristics.

**Supplementary Materials:** The following are available online at https://www.mdpi.com/article/10.3390/app11125687/s1. Figure S1: SEM micrograph of: (a) Sf-NPAM membrane surface and (b) Ox-NPAM membrane surface. Figure S2: Scheme of electrochemical test-cell. Figure S3: Membrane potential vs. KCl solutions concentration ratio for samples: Sf-NPAM, Ox-NPAM, Sf-NPAM+TiO$_2$ and Ox-NPAM + TiO$_2$ membranes, ideal anion-exchange membrane (solid line, t$_{Cl^-}$ = 1). Table S1: Atomic concentration percentages of the elements found on both surfaces of the studied samples

**Author Contributions:** L.G. and J.B. performed electrochemical characterizations and A.L.C. was responsible for optical characterization. Experiments were coordinated by J.B. All authors have read and agreed to the published version of the manuscript.

**Funding:** This research received no external funding.

**Informed Consent Statement:** Informed consent was obtained from all subjects involved in the study.

**Acknowledgments:** We thank V.M. de la Prida and V. Vega, University of Oviedo, Spain, for kindly supplying studied samples.

**Conflicts of Interest:** The authors declare no conflict of interest.

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
