# Peer review of "Influence of Pore-Size/Porosity on Ion Transport and Static BSA Fouling for TiO2-Covered Nanoporous Alumina Membranes"

_applsci, doi:10.3390/app11125687_

Round 1

Reviewer 1 Report

1. Page 2: although collaborators provided the membrane samples, it is highly recommended that the authors provide morphological images (e.g., SEM) of the membrane used in the study before/after the TiO2 ALD process.

2. Fig. 1, Page 5: the authors should provide a clear illustration.

3. Fig. 2, Page 5: What is the Debye length of the ion solution tested in this study? Also, the authors should explain how the electric double layer at each ion concentration affects ion transport properties of the TiO2 membranes as a function of charge (or pH). For example, the surface charge type/density of TiO2 can be varied at different pH.

4. In addition to the pore size and porosity of the membranes, authors also should discuss the effect of the surface charge on the results of the fouling test. From the atomic composition of the two membranes (Table SI-1), the surface properties (e.g., surface charge) of the membranes could be different.  

5. I can’t find any supporting information except Table SI-1.

Author Response

  1. Page 2: although collaborators provided the membrane samples, it is highly recommended that the authors provide morphological images (e.g., SEM) of the membrane used in the study before/after the TiO2 ALD process.

We totally agree with the reviewer indication that SEM images of the studied samples are important information and they should be showed; consequently, they are now presented in the Experimental section (new Fig. 1) to support pore-size and porosity information, as a way to avoid their consideration as paper results.

                        (a)                                    (b)

Figure 1: SEM micrograph of: (a) Sf-NPAM/TiO2 membrane surface and (b) Ox-NPAM/TiO2 membrane surface (from ref. [36]).

  1. Fig. 1, Page 5: the authors should provide a clear illustration.

We have tried to improve the scheme shown previous Fig.1 (Fig. 2 in this new version).

  1. Fig. 2, Page 5: What is the Debye length of the ion solution tested in this study? Also, the authors should explain how the electric double layer at each ion concentration affects ion transport properties of the TiO2 membranes as a function of charge (or pH). For example, the surface charge type/density of TiO2 can be varied at different pH.

The Debye length expression for non-confined symmetrical electrolytes can be easily obtained by the expression:  k-1 = (ereoRT/2FC)1/2, with er ~ 80, but it seems that reduced er values should be considered in case of nanoconfined situations (pore size lower than 100 nm) according to different author: Science 2018, 360, 1339-1342; RSC Adv. 2020, 10, 8628-8635; J. Phys. Chem. B 2001, 105, 5106–5109; J. Molecular Liquids 2021, 324, 115139.

Since the electrochemical measurements in this work were performed at a constant pH, we have not considered such effect in this analysis (a range between 6.5 and 7.5 is usually considered for isoelectric point of TiO2 nanoparticles used in many membrane modification, which significantly dependent on fabrication/modification conditions). However, in the Introduction of this new version we have included a paragraph with references indicating diverse changes associated to TiO2 membrane modification which include hydrophilicity and surface charge as well as the pH dependence of this latte, including a rather complete theoretical and experimental study (Croat. Chem. Acta 2006, 79, 95-106.).

On the other hand, modifications with TiO2 nanoparticles of different types of membranes for diverse applications (membrane bioreactors, oily emulsion filtration,…) seems to reduce fouling effect independently of polymeric membrane material and TiO2 particles immobilization procedure (entrapped, deposited or deep coating) [23-25]. In this context, the increase in the hydrophilic character of the TiO2-modified membranes (polymeric and ceramics) has been indicated as one of the key factors for membrane fouling reduction caused by filtration of oily wastewater or BSA static fouling, although other factor as TiO2 surface charge, which depends on solution pH and salt concentration, should also be considered [26-28].

  1. In addition to the pore size and porosity of the membranes, authors also should discuss the effect of the surface charge on the results of the fouling test. From the atomic composition of the two membranes (Table SI-1), the surface properties (e.g., surface charge) of the membranes could be different.  

Taking into account the reviewer comment, correlation between effective fixed charge concentration (Xef) and surface charge density (ss) proposed by Szymczyk et al (J. Membr. Sci. 2005, 253, 77) for membranes with cylindrical pores, ss = (Xef.F.rp/2), is included and used in this new version for value estimation. Table SI-1 showing chemical composition of both NPAM/TiO2 samples by XPS analysis (where only slight differences related to manufacture/environmental contamination are detected) is now presented in connection with these results.        

On the other hand, a correlation between effective fixed charge concentration (Xef) and surface charge density (ss) for cylindrical pores was already proposed by the following expression [45]: ss = (Xef.F.rp/2), where F represents Faraday constant. Surface charge density values for the studied membranes are: 7.2 mC/m2 for Sf-NPAM/TiO2 sample and 7.5 mC/m2 for Ox-NPAM/TiO2 one. Since the electrical charge exhibited by most membranes in contact with aqueous solutions (polar medium) corresponds to the adsorption of charged species from solution on the pore walls and/or dissociation of specific ionic groups, the similarity obtained for surface charge density values seems to be in concordance with the similitude in membrane surface material determined from XPS analysis. 

  1. I can’t find any supporting information except Table SI-1.

We hope that Supporting Images are correctly included in this new version.

Main changes included in this revised version are indicated in red colour.

Reviewer 2 Report

I recommend the authors to rework their manuscript according to the comments below:

Please check English throughout the text (e.g. L16-17, instead of ‘ions transport’ it should be used the singular form, L375-376 etc)

The abstract should be reworked as the sequence of aim/material/methods/results is not straight-forward for the readers and difficult to follow.

Abbreviations in abstract are used without their definition first.

L16-18 are not clear.

Transmittance depends on the pore size/porosity combination, how does this characterization technique be employed for fouling characterization of such membranes? The authors only presented the results in abstract without mentioning the connection/relation between the two. Please also clarify the connection also in the Introduction section.

Please support the modification with TiO2 with results from literature /references and also add available results in literature for the same aim – fouling.

What was the point/advantage of performing the anodization for the fouling study instead of simply using commercial alumina membranes?

Please add info on thickness of Al discs.

The synthesis of NPAM/TiO2 is not clear: it is advised to clarify that the effect on the porosity is determined from the synthesis of NPAM while the modification with TiO2 is kept constant.

L110 and L111 present different characteristics for the same sample!

Please describe all parameters and their values in L135.

The applied potential for EIS is open circuit potential?

Please describe parameters and values in L165.

Please correct relation in L190.

Please clarify L192-194.

Please rewrite L219-220 for more clarity.

Please indicate the parameters investigated in fig 2b also in caption. It would be interesting to add also the membrane potential for Ox-NPAM/TiO2 with KCl and CaCl2 solutions.

Hws about the membrane results in the absence of TiO2? It is advised to add such data.

What about higher / lower pore size? In order to confirm results, higher pore size is advised to be included, so as to measure the increase in co-ion transport.

Please rework the figures, they appear elongated and the legend has very small font to be read.

The authors should support their claims (see L299-300) with references. The effect of BSA is complex and the authors make a lot of assumptions.

The Fig 8 should include also non-coated NPAMs  to confirm assumptions in L361 and support also effect of fouling on light transmission.

Author Response

I recommend the authors to rework their manuscript according to the comments below:

Please check English throughout the text (e.g. L16-17, instead of ‘ions transport’ it should be used the singular form, L375-376 etc)

Thank you very much to reviewer for detection and indication of English mistakes, we hope they are been corrected in this new version.

The abstract should be reworked as the sequence of aim/material/methods/results is not straight-forward for the readers and difficult to follow.

Abbreviations in abstract are used without their definition first.

L16-18 are not clear.

According to reviewer indication we have rewritten the Abstract for clarification and identification purposes.

Abstract: The influence of geometrical parameters (pore radii and porosity) on ion transport through two almost ideal nanoporous alumina membranes (NPAMs) covered with a thin TiO2 layer by atomic layer deposition technique (Sf-NPAM/TiO2 and Ox-NPAM/TiO2 samples) is analyzed by membrane potentials and electrochemical impedance spectroscopy measurements. These results show the significant effect of pore radii (10 nm for Sf-NPAM/TiO2 and 13 nm for Ox-NPAM/TiO2) when compared with porosity (9 % and 6 %, respectively). Both electrochemical techniques were also used for estimation of protein (bovine serum albumin or BSA) static fouling and the results seem indicate deposition of BSA layer on the Sf-NPAM/TiO2 fouled membrane surface but pore-wall deposition in the case of Ox-NPAM/TiO2 fouled one. Moreover, a typical and simple optical technique such as light transmission/reflection (wavelength ranging between 0 and 2000 nm) was also used for membranes analysis, showing only slight transmittance difference in the visible region when both clean membranes are compared, but rather significant transmittance reduction (~ 18 %) was obtained for fouled Sf-NPAM/TiO2 sample respect to Ox-NPAM/TiO2, fouled one, which is associated to BSA deposition on membrane surface, thereby supporting electrochemical analysis results.

Transmittance depends on the pore size/porosity combination, how does this characterization technique be employed for fouling characterization of such membranes? The authors only presented the results in abstract without mentioning the connection/relation between the two. Please also clarify the connection also in the Introduction section.

We agree with the reviewer indication that transmittance depends on the pore size/porosity or “free volume (fv)”, which is not so different for the studied samples (fvOx-NPAM+TiO2/fvSf-NPAM+TiO2 = 1.13, which is now indicated in the text), but also on membrane material transparency. We have tried to clarify this latter point in this new version by including a new Figure presenting transmittance results for three alumina-based membranes, two of them  with similar pore-size (10 nm) and porosity (9 %) obtained covering the same Sf-NPAM with a layer of two different materials (Fe2O3 or Al2O3, samples Sf-NPAM/Fe2O3 or Ox-NPAM/Al2O3 respectively), which shown significant transmission differences (and band gap) in the visible region, while the other one is the Ox-NPAM sample covered with a layer of SiO2 (13 nm pore radii and 6 % porosity, sample Ox-NPAM/SiO2), where a higher level of transparency and differences in both the visible and infrared regions can be observed. 

Please support the modification with TiO2 with results from literature /references and also add available results in literature for the same aim – fouling.

References on the effect of TiO2 modification in different kinds of membranes are included in this new version.

What was the point/advantage of performing the anodization for the fouling study instead of simply using commercial alumina membranes?

Although commonly used commercial alumina membranes (e.g. AnoporeTM filters) are also obtained by anodization process, they do not show perfect ideal structure and even asymmetry, which significantly difficult transport modelling and parameters calculation, affecting different physicochemical parameters (including light transmission), Differences can be observed in next figure from Appl. Sci. 202010(14), 4864.

        (a)                                                           (b)

SEM Images of surface and cross-section of: (a) an experimental membrane; (b) a commercial AnoporeTM membrane, both obtained by electrochemical method.

Please add info on thickness of Al discs.

Al disc thickness, which is now indicated in this new version of the paper, is 0.5 mm.

The synthesis of NPAM/TiO2 is not clear: it is advised to clarify that the effect on the porosity is determined from the synthesis of NPAM while the modification with TiO2 is kept constant.

Pore radii (rp) and interpore distance (Dint) of nanoporous alumina membrane (NPAM) depend on electrolyte solution and voltage used in sample fabrication, and porosity (Q) is related to both parameters by: Q = (2p/Ö3)(rp/Dint)2) [Masuda, H.; Fukuda, K. Science 1995, 268, 1466-1468]. TiO2 layer coverage reduces pore radii and, consequently porosity, but it does not affect Dint value. We hope that this point is clarified in this new version.

The nanoporous alumina membranes (NPAMs) used as supports in this study were synthesized by the electrochemical two-step anodization method using high purity aluminum discs (Al 99.999 %, Goodfellow (UK); 0.5 mm thickness). Two different aqueous electrolyte solutions and anodization voltages were used in order to have membranes with different pore size and interpore distance: 0.3 M solution of sulfuric acid and applied voltage of 25 V for Sf-NPAM sample, and 0.3 M solution of oxalic acid and 40 V in the case of the Ox-NPAM sample; detailed information on this process is given in the literature [1, 3, 35]. In Supplenentary Information scanning electron microscope (SEM) pictures of both membrane surfaces is presented as Fig. SI-1.

Modification of Sf-NPAM and Ox-NPAM surfaces, both external and internal (pore-wall), with a thin TiO2 coated layer by atomic layer deposition (ALD) technique was performed in a Savannah 100 thermal ALD reactor from Cambridge Nanotech (Waltham, MA, USA), using high purity argon as carrier gas. Two precursors, titanium tetraisopropoxide (metal precursor) at 75 ºC and water (for substrate functionalization and as oxidant agent) at 60 º C, were used for the deposition of the conformal coating, as explained in a previous work [20]. These nanoporous alumina-based membranes will hereafter be called Sf-NPAM/TiO2 and Ox-NPAM/TiO2 respectively. Almost total TiO2-coverage of both alumina supports was already determined in a previous paper [36] by XPS technique, and the atomic concentration percentage (A.C. %) of the different elements found on the surfaces of each membrane is shown as Supplementary Information (Table SI-1); only superficial slight differences associated to manufacture/environmental contamination (sulphur detection for Sf-NPAM/TiO2 or excess of carbon in the case of Ox-NPAM/TiO2 sample) were determined. The thickness of the TiO2 layer was also determined (from depth-profile XPS analysis [36]), obtaining a value of ~ 6 nm, which agrees quite well with published results [21]. Fig. 1 shows SEM surface micrographs of both NPAM/TiO2 samples, and the following values for average pore-radii (rp), interpore distance (Dint) and porosity (Q = (2p/Ö3)(rp/Dint)2) [1]), weredetermined by surface analysis of SEM micrographs [36]: <rp> = 10 nm, <Dint> = 65 nm and <Q> = 9 % for Sf-NPAM/TiO2 sample, while <rp> = 13 nm, <Dint> = 105 nm and <Q> = 6 % for Ox-NPAM/TiO2, having both membranes a thickness of ~ 65 mm. Taking into account geometrical parameter values, rather similar theoretical hydraulic permeability is determined for both TiO2-covered membranes [9]: LHSf-NPAM/TiO2 = 1.92x10-12 m/s.Pa and LHOx-NPAM/TiO2 = 2.17x10-12 m/s.Pa. It should be mention that both NPAMs and NPAM/TiO2 samples were obtained in the Unidad de Membranas Nanoporosas, Universidad de Oviedo (Spain) and kindly submitted by Prof. V. de la Prida and Dr. V. Vega.

                        (a)                                               (b)

Figure 1: SEM micrograph of: (a) Sf-NPAM/TiO2 membrane surface and (b) Ox-NPAM/TiO2 membrane surface (from ref. [36]).

L110 and L111 present different characteristics for the same sample!

We thank very much to the reviewer the detection of this typing mistake, which is a key point for a better understanding of the paper!!! In this new version, we have modified and clarified this part.

Please describe all parameters and their values in L135.

This part has been rewritten as follows:

Membrane potentials (DFmbr), or equilibrium electrical potential difference between two NaCl solutions of different concentration (Cf =0.01 M and Cv ranging between 0.002 M and 0.1 M) at both membrane sides, were determined by measurements performed in the dead-end test cell shown already indicated (Supplementary Information, Fig. SI-2), with the two Ag/AgCl reversible electrodes (to Cl- ion) connected to a digital voltmeter (Yokohama 7552, 1GW input resistance) and the magnetic stirrers placed in the bottom of each cell working at a stirring rate of 540 rpm (to minimize the concentration-polarization at the membrane surfaces [37]). Since measured values (DE) include two different contributions, electrode potential (DFelect) and membrane potentials (DFmbr), these latter values were obtained by subtracting to each DE value that corresponding to DFelect (which depends on electrolyte concentrations, DFelect = (RT/zF)ln(Cv/Cf), where R and F are gas and Faraday constants, respectively, while T represents the temperature of the system), then: DFmbr = DE - DFelect [38]. Measurements were performed at standard pH (5.8 ± 0.3) and room temperature (25 ± 2) ºC.

The applied potential for EIS is open circuit potential?

Yes, EIS measurements are open circuit potential, which is now indicated in the text.

Please describe parameters and values in L165.

This part is now written as follows:

The expressions for Donnan and diffusion potentials are [38]:

Donnan potential, ∆øDon = (RT/F)ln[(wXef/2C) + [(wXef/2C)2+1)1/2]]               (1)

Diffusion potential, ∆fdif = - (RT/F)[(t_ - t+)]ln(Cv/Cf) = (RT/F)[(1 - 2t-)]ln(Cv/Cf)   (2)

with w = -1/+1 for negatively/positively charged membranes, while Ci indicates solution concentration (i = v for variable solution concentration and i = f for fixed solution concentration), and R, F and T have been previously identified. According to the Teorell-Meyer-Sievers model [41-42], membrane potential consists in the sum of two Donnan potentials (one for each membrane/solution interface) and the diffusion potential in the membrane.

Please correct relation in L190. Please clarify L192-194.

That paragraph has also been changed:

Since ion transport numbers not only depend on membrane charge but electrolyte electrochemical parameters, membrane ionic permselectivity (PSi) or relative variation of counter ions in the membrane with respect to their values in solution use to be commonly considered; then, for positively charged membranes and NaCl solutions, PSCl-, is expressed as [43]: PSCl- = (tCl- − toCl)/toNa+, where tCl- is the transport number of the counter-anion in the membrane pores, while toCl- and toNa+ represent the transport numbers of the anion and cation, respectively, in the NaCl solution.

Please rewrite L219-220 for more clarity.

New paragraph redaction:

The fit of the experimental values can be performed taking into account both Donnan and diffusion potentials as well as membrane electroneutrality, and it allows the determination of effective membrane fixed charge concentration (Xef), anion/cation diffusion coefficients ratio (DCl-/DNa+, since (tCl-/tNa+) = (DCl-/DNa+) [41]) or anion permselectivity, as it has already been explained in previous papers [42-44].

Please indicate the parameters investigated in fig 2b also in caption. It would be interesting to add also the membrane potential for Ox-NPAM/TiO2 with KCl and CaCl2 solutions.

Comparison of membrane potential with KCl solutions for Sf-NPAM/TiO2 and Ox-NPAM/TiO2 samples is presented in Supplementary Information (Fig. SI-2).

According to referee indication fitted values shown in Fig. 2b caption.

Hws about the membrane results in the absence of TiO2? It is advised to add such data.

Comparison of membrane potential for Sf-NPAM and Ox-NPAM samples is also presented in Supplementary Information (Fig. SI-2).

Combination of both membrane geometry and surface material can be observed in Supplementary Information (Fig. SI-3) where a comparison of membrane potential values measured with KCl solutions for Sf-NPAM, Ox-NPAM, Sf-NPAM/TiO2 and Ox-NPAM/TiO2 is presented [36], while the effect of pore size, porosity and structure asymmetry for NPAMs has already been reported [35, 45].

Figure SI-3: Membrane potential vs KCl solutions concentration ratio for samples: Sf-NPAM (o), Ox-NPAM (), Sf-NPAM+TiO2 (●) and Ox-NPAM+TiO2 () membranes; ideal anion-exchange membrane (solid line, tCl- = 1) (from ref. [36]).

What about higher / lower pore size? In order to confirm results, higher pore size is advised to be included, so as to measure the increase in co-ion transport.

As it was indicated in our previous answer, Fig. SI-2 in Supplementary Information sows the effect of pore-size by comparing Sf-NPAM and Ox-NPAM, as well as Sf-NPAM/TiO2 and Ox-NPAM/TiO2. Moreover, references on pore size, porosity and membrane structure effect on ion transport are also indicated [35,45].

Please rework the figures, they appear elongated and the legend has very small font to be read.

In this new version we have treated to improve the size of the figures by using a unique column and two different lines, but typography and size (both text and legend) depend on journal format.

The authors should support their claims (see L299-300) with references. The effect of BSA is complex and the authors make a lot of assumptions.

Although we agree with the complexity of BSA fouling, we think the electrochemical results support our interpretation on membrane fouling, taking into account that “static” fouling (without hydrostatic pressure forcing the BSA solution through the nanopores) is involved, and considering membrane pore-size and protein stokes radius. A reference on the use of optical technique (SE) for human serum albumin by changes in refraction index is also indicated.

The Fig 8 should include also non-coated NPAMs to confirm assumptions in L361 and support also effect of fouling on light transmission.

Next figure shows the comparison of transmittance spectra for Ox-NPAM (black line) and Ox-NPAM+SiO2 (cyan line) samples where the effect of both geometrical parameters and material are included; we has included in this new version of the work the previous figure for better identification of each effect. 

We actually thank the reviewer for English/typing mistakes indications. We hope they have been corrected in this new version of the paper.

Round 2

Reviewer 1 Report

This manuscript is recommended for publication since authors responded thoroughly to the comments.

Reviewer 2 Report

The authors have addressed the review comments satisfactorily.